# Gene–Environment Interaction: Small Deletions (DELs) and Transcriptomic Profiles in Non-Melanoma Skin Cancer (NMSC) and Potential Implications for Therapy

**DOI:** 10.3390/cells14020095

**Published:** 2025-01-10

**Authors:** Farzana Jasmine, Armando Almazan, Yuliia Khamkevych, Maria Argos, Mohammad Shahriar, Tariqul Islam, Christopher R. Shea, Habibul Ahsan, Muhammad G. Kibriya

**Affiliations:** 1Institute for Population and Precision Health (IPPH), University of Chicago, Chicago, IL 60637, USA; farzana@uchicago.edu (F.J.); armando.almazan@bsd.uchicago.edu (A.A.); yuliiak@uchicago.edu (Y.K.); mhshahriar@bsd.uchicago.edu (M.S.); hahsan@bsd.uchicago.edu (H.A.); 2Department of Environmental Health, School of Public Health, Boston University, Boston, MA 02118, USA; argos@bu.edu; 3UChicago Research Bangladesh (URB), University of Chicago, Dhaka 1230, Bangladesh; tariqulislam@urb-bd.org; 4Division of Dermatology, Department of Medicine, University of Chicago, Chicago, IL 60637, USA; cshea@bsd.uchicago.edu; 5Department of Public Health Sciences, Biological Sciences Division, University of Chicago, Chicago, IL 60637, USA

**Keywords:** non-melanoma skin cancer, basal-cell carcinoma, somatic mutation, small deletion, short tandem repeat, arsenic, gene–environment interaction, immune checkpoint inhibitors, *IL-17* signaling, *TGF-β* signaling

## Abstract

Arsenic (As) is a risk factor for non-melanoma skin cancer (NMSC). From a six-year follow-up study on 7000 adults exposed to As, we reported the associations of single-nucleotide variation in tumor tissue and gene expression. Here, we identify the associations of small deletions (DELs) and transcriptomic profiles in NMSC. Comparing the (a) NMSC tissue (*n* = 32) and corresponding blood samples from each patient, and (b) an independent set of non-lesional, healthy skin (*n* = 16) and paired blood, we identified NMSC-associated DELs. Differential expressions of certain gene pathways (*TGF-β* signaling pathway, *IL-17* pathway, *PD-L1* pathway, etc.) showed significant interactions with these somatic DELs and As exposure. In low-As-exposure cases, the DELs in *APC* were associated with the up-regulation of inflamed T-Cell-associated genes by a fold change (FC) of 8.9 (95% CI 4.5–17.6), compared to 5.7 (95% CI 2.9–10.8) without *APC* DELs; in high-As-exposure cases, the *APC* DELs were associated with an FC of 5.8 (95% CI 3.5–9.8) compared to 1.2 (95% CI −1.3 to 1.8) without *APC* DELs. We report, for the first time, the significant associations of somatic DELs (many in STR regions) in NMSC tissue and As exposure with many dysregulated gene pathways. These findings may help in selecting groups of patients for potential targeted therapy like PD-L1 inhibitors, IL-17 inhibitors, and TGF-β inhibitors in the future.

## 1. Introduction

Human skin is composed predominantly of keratinocytes along with Merkel cells and pigment-producing cells (melanocytes). Exposure to arsenic (As) is associated with the development of keratinocytic carcinoma, also known as non-melanoma skin cancer, a term that covers squamous-cell carcinoma in situ or Bowen’s disease (BD), basal-cell carcinoma (BCC), and invasive squamous-cell carcinoma (SCC). Previously, we reported the association of frequently found single-nucleotide variations (SNVs) in tumor tissue and gene expression in an As-exposed Bangladeshi cohort [1]. Here, we identify the associations of small deletions and transcriptomic profiles in NMSC with high and low As exposure. The Global Burden of Disease Cancer Collaboration in 2017 showed that the annual incidence rate of NMSC was 7664, with an age-standardized incidence rate (ASIR) of 65 deaths per 100,000 [2]. Data obtained from the Singapore Cancer Registry from 1968 to 2016 reported cases of BCC and SCC to be 8367 and 3598, respectively [3]. A Japanese cancer registry reported 67,867 skin cancer incidents from 2016 to 2017, among which the percentages of BCC and SCC were 37.2% and 43.9% (of which 18.3% were in situ), respectively [4]. In Bangladesh, we followed up with 7000 adults for 6 years who were exposed to As. During this follow-up period, 2.2% of the males and 1.3% of the females developed BCC, while 0.4% of male and 0.2% of female participants developed SCC [1,5].

As exposure has been tied to skin cancer and other malignancies worldwide [6,7]. In 2012, the International Agency for Research on Cancer (IARC) working group reported that the strongest evidence for the association of human cancer with arsenic in drinking-water came mainly from studies in five countries—Taiwan, Chile, Argentina, Bangladesh, and India [8]. The characteristic arsenic-associated skin tumors include SCC, BD, and BCC [8]. In 2013, findings from a US-based prospective cohort study also suggested the role of low–moderate As exposure in the development of lung, prostate, and pancreatic cancer [9]. The As methylation capacity of the host may have a role in the development of arsenic-induced skin disorders [10].

A large study, including an As-exposed cohort from Health Effects of Arsenic Longitudinal Study (HEALS, *n* = 2434), Strong Heart Study (SHS, NCT00005134, *n* = 868), and New Hampshire Skin Cancer Study (NHSCS, *n* = 666), revealed the association of a variant in the *AS3MT* gene and the percentage of dimethylarsinic acid (DMA%) in urine [11]. Another study on HEALS participants identified a protein-altering variant in *FTCD* (rs61735836) that was associated with both As metabolism efficiency and risk of As-induced skin lesions [12].

The mechanisms by which As induces cancer are unclear. Evidence has suggested that the carcinogenesis from the metal is derived from its ability to cause epigenetic changes [13]. It is known that the inactivation of *PTCH1* and the up-regulation of the hedgehog (Hh) signaling pathway are most likely pivotal events in As-related pathogenesis [14]. Somatic mutations in *PTCH1*, *SMP*, *SUFU*, *NRAS*, *HRAS*, *KRAS*, *PIK3CA*, *RAC1*, *FBXW7*, *RB1*, *CDKN2A*, *NOTCH1*, *NOTCH2*, *CASP8*, and *ARID1A* have been reported [15,16,17]. All of these studies report SNVs. To our knowledge, no study has focused on small deletions (DELs). We have previously seen that sunlight exposure is associated with a large number of somatic mutations in different genes in non-lesional, apparently healthy skin [1]. Therefore, identifying a DEL in NMSC samples does not necessarily establish the association between that DEL and the NMSC pathogenesis. In this study, we (a) identified NMSC-associated DELs found only in NMSC and not healthy skin and (b) examined if such DELs were associated with the dysregulation of gene pathways relevant to pathogenesis and/or therapeutic options in NMSC.

## 2. Materials and Methods

### 2.1. Study Population and Biological Samples

For this study, we used the same participants who were included in a previous study, where we focused on somatic SNVs in NMSC [1]. The participants were part of a large-scale double-blind, placebo-controlled study, the Bangladesh Vitamin E and Selenium Trial (BEST, NCT00392561) [18]. In this study, as cases, we included the first 32 participants, developing biopsy-proven NMSC during the prospective follow-up. We also collected non-lesional or apparently healthy skin tissue surrounding the margin of arsenical keratosis lesions from 16 independent patients. These samples were used as normal controls. All the biopsy tissue samples were preserved in RNA stabilizing buffer (RNAlater, Thermo Fisher Scientific, Waltham, MA, USA) and were stored at −86 °C in a freezer until nucleic acid extraction. We also collected spot urine samples for the measurement of urinary As–creatinine ratio (UACR) at baseline as a measure of As exposure. For both the cases (*n* = 32) and the controls (*n* = 16), we also collected whole blood samples in EDTA tubes from the same individuals as a source of germline DNA. Sequencing from the tissue DNA was compared against the corresponding blood DNA from the same individual to detect somatic mutations. Thus, we sequenced 96 DNA samples (32 tissue DNA and 32 blood DNA for cases; 16 tissue DNA and 16 blood DNA for controls). For the gene expression study, we used 48 RNA samples from 32 NMSC tissues and 16 non-lesional skin tissues for RNA sequencing. For the patient characteristics, please see Appendix A.

### 2.2. Arsenic Exposure Measurement

The urinary total As concentration and urinary creatinine were measured by inductively coupled plasma mass spectrometry [19] and by a colorimetric method, respectively [20]. The log_2_-transformed UACR showed a strong correlation to the log_2_-transformed well water As concentration (r = 0.66) [21].

### 2.3. Somatic Mutation Assay

For somatic mutation, we used a targeted amplicon sequencing approach. A total of 409 genes (see Appendix A for total gene list) were sequenced using the AmpliSeq for Illumina comprehensive Cancer (Illumina Inc., San Diego, CA, USA). Tumor and non-lesional, apparently healthy skin tissue DNA were compared to corresponding blood DNA from same individuals to detect somatic mutations. In this manuscript, for the non-lesional, apparently healthy skin tissue, we use the term “healthy skin tissue”. Sequencing was performed on the Illumina HiSeq platform (San Diego, CA, USA).

### 2.4. Gene Expression Assay

For RNA sequencing, we used the Lexogen Quantiseq 3′ mRNA–Seq kit (Vienna, Austria) for library preparation following the manufacturer’s protocol as described in our earlier paper [1]. Sequencing was performed on the Illumina HiSeq platform (San Diego, CA, USA).

This study was approved by the Institutional Review Board of The University of Chicago Medicine protocol code IRB19-0724, approved on 24 September 2019.

### 2.5. Mutation Detection

For somatic mutation detection from the Illumina sequencing data, we used the Targeted Amplicon Sequencing (TAS) protocol using the CLC genomics Workbench23 (https://digitalinsights.qiagen.com/; accessed on 1 November 2024). The in-built workflow removed the germline variants found in the public database (db SNP, 1000 genomes project, dbSNPs common, and hapmap) that were also found in the mapped reads. Variants outside the target region were also removed, as they are likely to be false positives due to non-specific mapping of sequencing reads. The parameters for the low-frequency variant detection were set at a minimum coverage of 10, minimum count of 2, and minimum frequency of 2%. We used variant calling quality score of Q60 as the cut-off for the list of somatic mutations.

### 2.6. Statistical Methods

For transcriptomic data processing, we used Partek Flow (version 10.0) (https://www.partek.com/partek-flow/, accessed on 11 November 2022) using the STAR aligner for alignment, and the final gene count data were expressed as counts per million reads (CPM). Log_2_-transformed CPM data were used for ANOVA and Gene-set ANOVA as described in the previous paper [1,5]. To compare the magnitudes of the differential expression of “Gene set” in the absence or presence of a factor (DEL in this paper), we included interaction terms in the model as shown below:Model: Y = μ + T + G + TxG + TxMut + ε
where Y represents the expression status of a gene set category, μ is the common effect or average expression of the gene set category, T is the tissue-to-tissue (tumor/non-lesional) effect, G is the gene-to-gene effect, TxG is the differential pattern of gene expression in different tissue type, TxMut is the interaction term, and ε is the random error. ANOVA-generated *p*-values and the corresponding F-ratios were reported. We also report the Bonferroni correction that was used for multiple testing.

## 3. Results

In tissue samples from the first 32 patients with NMSC, sequencing 409 cancer-related genes revealed a total of 6594 somatic mutations in the NMSC tissue (representing unique 3277 genomic coordinates) compared to germline DNA samples from the same patients. In the same way, in healthy skin tissue from 16 individuals, we detected 2454 somatic mutations (representing unique 1434 genomic coordinates) compared to germline DNA from the same individuals. We divided the somatic mutations into single-nucleotide variants (SNVs), insertions (INS), and deletions (DELs). Previously, we reported the results from the SNVs and their functional effects [1]. Here, we present the DELs and their functional effects. Figure 1A shows the overlap between the list of somatic DELs in NMSC tissue (a total of 2694 DELs covering 1077 unique genomic coordinates in 32 samples) and DELs detected in healthy or healthy skin tissue (a total of 1094 DELs covering 605 unique genomic coordinates in 16 samples). Figure 1A shows the overlap by unique genomic coordinates, and Figure 1B shows the overlap by unique genes. It may be noted that there are a large number of genes (*n* = 213) that may harbor somatic mutations (DELs in this case), even in apparently healthy skin tissue, as well as in NMSC. A smaller number of genes are mutated only in NMSC tissue (*n* = 71), and a similarly small number of genes (*n* = 15) are mutated in healthy skin that are not mutated in NMSC tissue.

The Venn diagram identified three groups of somatic DELs, listed as follows:NMSC-associated DELs: A total of 965 DEL events representing 617 unique genomic coordinates (light pink region of the Venn diagram in Figure 1A).DELs common in NMSC and healthy skin: A total of 1729 DEL events representing 460 unique genomic coordinates (light green and light pink overlapping region in Figure 1A).DELs associated with healthy skin: A total of 170 DEL events representing 145 unique genomic coordinates (light green region in Venn diagram in Figure 1A).

Figure 2A–C show the top 40 genes in each of the above three groups. It may be noted that if we look at a single gene, for example, *BRAF*, there are some DELs in *BRAF* that may be found in healthy skin tissue; DELs in some other location of *BRAF* are only found in NMSC, and yet other DELs in *BRAF* may be found in both NMSC and healthy skin tissue. Therefore, the identification of tumor-specific DELs or mutations is important.

Then, from the NMSC-associated DELs, we identified BCC-associated DELs (found only in BCC cases but not in healthy skin tissue) among the 26 patients with BCC and SCC-associated DELs (found only in SCC cases but not in healthy skin tissue) among the 6 patients with SCC (see Figure 3). One gene may have multiple BCC-associated DELs. If an individual patient had any of the BCC-associated DELs in a particular gene (e.g., *MTR*), we considered the patient as having *MTR* DEL. The percentage of patients showing BCC-associated DELs in a particular gene is also shown on the right side of Figure 3. For example, 50% of the patients with BCC (13 out of 26) had BCC-associated DELs in the *MTR* gene. Considering the fact that we had only six patients with SCC, for further analysis we considered only the twenty-six patients with BCC. The number of BCC tissue samples and healthy skin tissue samples with or without BCC-associated DELs in frequently mutated genes is also shown in Appendix A.

### 3.1. BCC-Associated DELs

It was interesting to note that out of the total 965 BCC-associated DEL events among 26 patients with BCC (representing 617 unique genomic coordinates), more than two-thirds (663 events in 404 unique genomic coordinates) were found in the homopolymeric region of the genome. These represent the simplest of simple sequence repeats [22]. The number of repeats ranged from 3 to 25. In the next step, we examined whether the frequently encountered DELs had any functional effect on the expression of different gene pathways. In the Geneset ANOVA models, we included the interaction term “Tissue (0 = healthy, 1 = NMSC) × DEL (0 = absence of DEL, 1 = presence of DEL)” to evaluate whether the magnitude of differential expression (tumor vs. healthy) was significantly different if the NMSC tissue sample had a DEL or not. We examined all the KEGG pathways. The list of *APC* DELs is shown in Table 1. It may be noted that one of the variants was a frameshift mutation. All of the DELs in *APC* were in the homopolymeric region representing the simplest form of a short tandem repeat (number of repeat “A”s or “T”s ranging from 3 to 13).

The detailed associations of differential expression of these KEGG pathways with BCC-associated DELs in the *APC* gene are presented in Appendix A. It may be noted that the magnitude of overexpression of the genes involved in the *PD-L1* expression and *PD-1* checkpoint pathway was more marked (interaction *p* = 5.2 × 10^−21^) in patients with BCC who had BCC-associated DELs in *APC* [FC = 2.9 (95% CI 2.5–3.4); see Figure 4B] compared to patients without *APC* DELs [FC = 1.4 (95% CI 1.2–1.6); see Figure 4A]. This result suggests that perhaps patients with BCC who have *APC* DELs may be better candidates for immune checkpoint inhibitor (ICI) therapy compared to those without *APC* DELs.

Similarly, the magnitudes of overexpression of genes involved in the *TNF*-α pathway (see Appendix A), *TGF-β* signaling pathway (see Figure 5A,B), *NF-κB* signaling (see Appendix A), and *IL-17* signaling pathways in BCC were also significantly more in the presence of *APC* DELs (FC ~3) compared to cases where *APC* DELs were absent (FC ~1.5) (see Appendix A). These findings may suggest that TGF-β inhibitors, NF-κB inhibitors, or IL-17 inhibitors may lead to a better response in BCC patients with *APC* DELs.

The associations of differential expression of these KEGG pathways with BCC-associated DELs in *MLH1*, *MTR*, *DST*, and *MSH6* are presented in Appendix A, respectively. The presence or absence of DELs in these genes also showed a significant difference in magnitude of the differential expression of some gene pathways.

### 3.2. Gene–Environment Interaction

In our previous study, we documented how the magnitudes of differential expression of many of these gene pathways were affected by the level of As exposure [1]. Therefore, in the next step, we asked if the DELs in the genes mentioned in Section 3.1 (*APC*, *MLHI*, *MTR*, *DST*, and *MSH6*) had any gene–environment interaction. Considering that all of the patients in this study were exposed to As through drinking As-contaminated water from deep tube wells, we divided the patients into low As exposure (UACR ≤ 192 μg/g) and high As exposure (UACR > 192 μg/g) on the basis of median UACR for this population [5]. By gene, we mean the presence or absence of DELs in a particular gene, and by environment we mean low or high As exposure. Because of the limited number of patients, we only tested the five genes that were more frequently mutated (DELs). We tested the interactions for all KEGG pathways (for details, see Appendix A for *APC*, *MLHI*, *MTR*, *DST*, and *MSH6*, respectively). For many of the KEGG pathways, statistically significant interactions were seen (even after Bonferonni correction for multiple testing); however, considering clinically relevant potential targeted therapy, three of the KEGG pathways are shown in Table 2, where the differential expressions of KEGG pathways were checked in four sub-groups of patients.

In general, these analyses suggest a potentially meaningful interaction in a way that the overexpression of *TGF-β* signaling, *IL-17* signaling, and *PD-L1* expression were all more pronounced in patients with low As exposure having *APC* DELs. In other words, this sub-group of patients with BCC may potentially respond better to therapy targeting these pathways.

### 3.3. Inflamed T Cell Markers

In connection with the potential use of immune checkpoint inhibitors (ICIs), we also looked for gene expression of the inflamed T-cell-related genes. On average, the genes associated with inflamed T-cells were up-regulated by FC = 2 (95% CI 1.4–2.9) in the absence of DEL *APC*, whereas the genes were up-regulated by FC = 8 (95% CI 5.3–12.5) in the presence of DEL *APC* (interaction *p* = 3.4 × 10^−10^; see Figure 6A,B).

As exposure also interacts with the differential expression of the genes associated with inflamed T-cells (see Figure 6C,D). We also checked if there was interaction between *APC* DELs and As exposure (see Table 3). We noticed that the overexpression of genes related to inflamed T-cells was minimal in patients without *APC* DELs who had high As exposure compared to those with *APC* DELs and/or low As exposure. This result suggests that ICI therapy could be most effective in patients with low As exposure and/or *APC* DELs.

### 3.4. APC DELs and PTCH1 Mutation

The *PTCH1* gene is frequently mutated in BCC, and in our previous study we also showed the significant functional effect of non-synonymous SNV mutation in *PTCH1* in this population [1]. Therefore, we checked if the *APC* DELs and non-synonymous SNV in *PTCH1* co-occurred in the same patients. In fact, out of the 10 patients with *APC* DELs, 8 had ns-SNV in *PTCH1*, and out of the 12 patients with ns-SNV in *PTCH1*, 8 also had *APC* DELs (*p* = 0.01). Therefore, these two mutations may not be mutually exclusive. We checked the differential expression of genes related to inflamed T-cells among the sub-groups of patients by ns-SNV in *PTCH1* and *APC* DEL mutation status (see Table 4). The result suggests that patients with BCC without *APC* DELs and without ns-SNV in *PTCH1* had the least overexpression of the genes related to inflamed T-cells and are therefore least likely to respond to ICI therapy, whereas patients with *APC* DELs and/or ns-SNV *PTCH1* mutation have a significantly more pronounced overexpression of inflamed T-cells and may be more likely to respond to ICI therapy.

## 4. Discussion

In this study, we show that small deletions in many cancer-related genes can be found in skin tissues. Some of them are present in non-lesional, apparently healthy skin, some are found only in NMSC tissues, and some may be seen in both cases. Therefore, not all DELs may have implications in skin cancer. We tried to identify the NMSC-associated DELs (in both BCC and SCC) that may have implications. The case–control design of this study did not allow us to comment on the exact mechanism of carcinogenesis. However, by exploring the associations of these somatic changes (DELs) and the differential expression of gene and gene pathways (at the genome-wide level) in tissues from the same individuals, we could shed light on the pathogenesis of NMSC. The analysis also provides the molecular basis of potential utility of some groups of medications (ICI, IL-17 inhibitors, TGF-β inhibitors) in a selective subgroup of patients with BCC (patients with BCC-associated *APC* DELs exposed to lower levels of As).

In general, when we speak of somatic mutations in NMSC, the main focus of most of the studies is on SNV and not INS or DELs. In our previous paper [1], we reported the SNVs and gene–environment interaction in NMSC in this same patient population. Observing a large number of DELs that could be identified with high confidence in the same samples, we attempted to explore further. In fact, we found that these DELs show an association with the magnitude of dysregulation of multiple gene pathways that may have implications in translational medicine. To our knowledge, we report for the first time the effects of small DELs (involving a single base or a few bases) in BCC or SCC tumor tissue and specifically their interaction with As exposure. These findings are also interesting because (a) the majority of these DELs were detected in a homopolymeric region of the genome that represents the simplest form of short tandem repeat (STR), and (b) we could see association of these DELs with many dysregulated gene pathways. Very recently, the significance of STRs has been highlighted in the context of human diseases [22,23,24]. INS or DELs in such regions have been reported in germline DNA; however, to our knowledge, no human studies have shown the significance of “somatic mutation” in cancer tissue in such regions.

There are a few studies on cytogenetics in NMSC, but those detect gain or loss of relatively large genomic regions (typically > 500 K). The incidence of DNA aneuploidy in BCC is 9% to 40%, and in SCC it is estimated to be 25% to 80% [25]. In a review, cytogenetic changes were shown by karyotyping [26]. There is frequent gain in 6p, 6q, 9p, 18, and X chromosome in BCC and a loss of 9q. Similarly, SCC showed a frequent gain of 3q and 8q along with a loss of 3p, 8p, 9p, 13q, and 18q [26].

In general, there is evidence that immunotherapy may be associated with an increased risk of skin cancer. A systemic review was performed of 19 studies including a total of 13,739 patients with psoriasis who were treated with IL-17 inhibitors, IL-23 inhibitors, and Janus kinase (*JAK*) inhibitors. The overall incidence rate of melanoma was 0.08 (95% CI 0.05–0.15) events per 100 patient years, and the overall incidence rate of NMSC was 0.45 (95% CI 0.33–0.61) events per 100 patient years in patients who received targeted therapy compared to the others [27]. However, some recent studies indicate the association of chronic inflammatory change and cancer development [28,29,30,31,32]. *IL-17* dysregulation may be a major pathogenic factor for cancer development. Ablation of *IL-17* reduces tumorigenesis in mouse models [28,33,34,35,36]. IL-17 inhibitors may (a) increase sensitivity to chemotherapy and radiation [32], (b) suppress metastasis [32], (c) improve the efficacy of immunotherapy, especially for tumors that are resistant to ICI [30], (d) or, in combination with ICI, may reduce the tumor burden in lung cancer [31,32].

Another meta-analysis revealed that the use of immunotherapy was linked to a greater risk of NMSC compared to no use of immunotherapy in three common inflammatory diseases. Amongst them, biological therapy increased the risk of NMSC in patients with rheumatoid arthritis (RA) (RR = 1.24, 95% CI 1.13–1.36) and psoriasis (RR = 1.28, 95% CI 1.07–1.52) but not the risk of those with IBD (RR = 1.49, 95% CI 0.46–4.91) [37].

One group investigated the efficacy of arsenic trioxide on the T-regulatory cell ratio and the levels of IFN-γ, IL-4, IL-17, and TGF-β1 in the peripheral blood of patients with severe aplastic anemia. The results showed that As significantly increased the proportion of T -regulatory cells (*p* < 0.05). In addition, As significantly reduced the levels of IFN-γ, IL-4, IL-17, and TGF-β1 [38]. The relationship between As exposure and the TGF pathway may be bidirectional [39]. As exposure up-regulated the expression of *TGF-β1*, *p-Smad2/3*, *α-SMA*, Collagen1/3, and fibronectin (FN). A previous study showed that at a low dose (≤5 µmol/L) As exposure can up-regulate *TGF-β1* expression, but a high dose of As caused down-regulation of *TGF-β1* [39]. This is along the same lines of our findings that indicated that the overexpression of *TGF-β* signaling pathway genes was less marked among patients exposed to higher levels of As compared to the patients exposed to lower levels of As.

A systematic review of 14 studies showed evidence that proteins like TGF-β, HBD, and cathelicidin play a role in developing BCC [40]. The importance of *PPAR-γ* signaling, *TGF-β* signaling, SHH, and *p53* in the pathogenesis of BCC are seen in other studies [41]. Activation of *Ras*/*MEK*/*ERK1*/2 and *TGF-β*/*Smad2* signaling leads to increased accumulation of laminin-332 and accelerated invasion [42]. A study showed that B-Raf inhibitor PLX8394 blocks *TGF-β* signaling in wild-type B-Raf and hyperactive Ras. It significantly inhibited the growth of human SCC tumors and in vivo collagen degradation in a xenograft model [43]. Overexpression of *TGF-β* promotes malignant progression and metastasis [44]. This is implemented by the epithelial to mesenchymal transition, in which cuboidal epithelial-like cells change to the elongated spindle and invasive phenotype. Mechanistic studies and clinical trials targeting *TGF-β* signaling have been reviewed [45].

PD-1 and PD-L1 are major immune checkpoint molecules. Cancer cells can use the PD-1/PD-L1 axis to cause immune escape in cancer development [46]. Some previous studies showed the over-expression of PDL1 in BCC tissue [47,48]. Our present study clearly suggests that the *PD-L1* pathway and genes related to inflamed T-cells are overexpressed in BCC. This overexpression was significantly more pronounced in the presence of *APC* DELs and especially in patients who had exposure to low level of As, making them potentially good candidates for ICI therapy. On the other hand, patients with high As exposure who did not have *APC* DELs showed lower levels of *PD-L1* dysregulation and are therefore less likely to benefit from ICI therapy.

In this study, our first objective was the detection of NMSC-associated DELs; the second step was to see if such DELs (at the gene level) were associated with different magnitudes of dysregulation in some gene pathways, which may help explain the biological and/or therapeutic relevance of molecular genomic data. In a previous study, we saw that the magnitudes of dysregulation of such gene pathways were associated with As exposure status. Therefore, in the third step of this current study, we asked if there was any interaction between these gene-level DELs and the As exposure status. We acknowledge the limitation of sample size in our study. We also want to point out that this study was not designed to detect all gene–environment interactions at a genome-wide scale addressing thousands of SNPs or all possible mutations, which is only possible in large-scale studies [49]. Considering this limitation, we restricted the GxE interaction to only a few genes that frequently harbored BCC-associated DELs.

All the patients involved in this study were treated with total excision of the lesion and did not receive or require any chemotherapy or targeted therapy. Currently, ICIs are used in advanced or metastatic BCC or that cannot be removed surgically or do not respond to hedgehog signaling pathway inhibitors. With the current findings from molecular genomics, in the future we should be able to select (a) a sub-group of patients who are more likely to respond and (b) a sub-group who are least likely to respond to ICI or targeted therapy like IL-17 or TGF-β inhibitors if such therapy is needed in addition to total excision.

## 5. Conclusions

For the first time, we report BCC- and SCC-associated small DELs in multiple genes (including many in STR regions) in tumor tissues that are associated with the dysregulation of many cancer-related gene pathways. We show that, in patients exposed to As, the magnitude of dysregulation also depends on the interaction of such DELs and the As exposure status. These findings may help in selecting groups of patients for potential targeted therapy like ICI, IL-17 inhibitors, and TGF-β inhibitors in the future.

## Figures and Tables

**Figure 1 cells-14-00095-f001:**
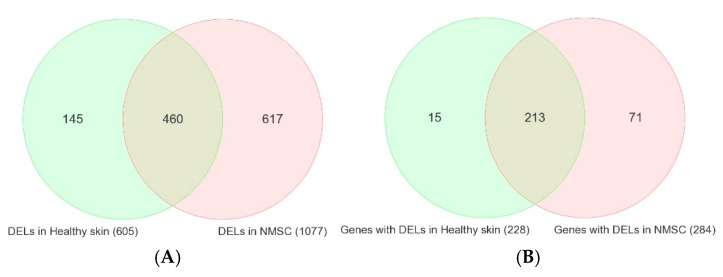
Venn diagram showing overlap between the DELs identified in healthy skin tissue (in light green) and NMSC tissue (in light pink). Overlap by unique genomic coordinates are on the left Venn diagram (**A**), and overlap by unique gene symbols are on the right Venn diagram (**B**).

**Figure 2 cells-14-00095-f002:**
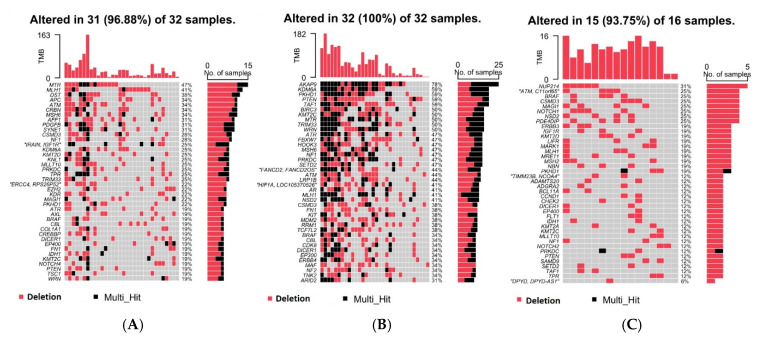
Top 40 genes with DELs found in (**A**) NMSC only, (**B**) common between NMSC and healthy skin, and (**C**) healthy skin only. Genes are shown in rows, while each column represents an individual patient.

**Figure 3 cells-14-00095-f003:**
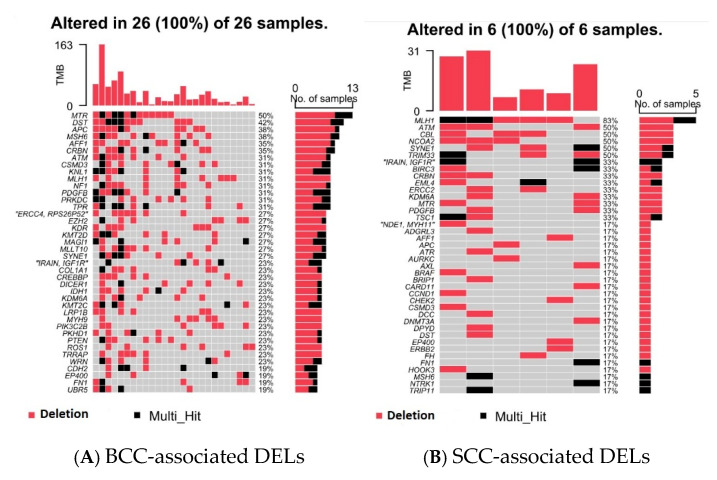
Top 40 genes harboring the BCC-associated DELs (**A**) and SCC-associated DELs (**B**) that are not found in healthy skin tissue. Genes are shown in rows, while each column represents an individual patient.

**Figure 4 cells-14-00095-f004:**
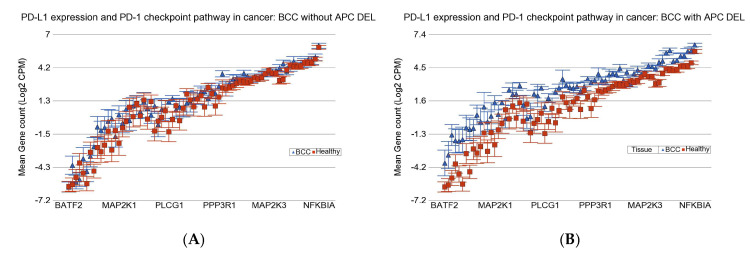
Differential gene expression of the *PD-L1* and *PD-1* checkpoint pathway in BCC tissue (in blue) compared to healthy skin tissue (in red). BCC tissues with no *APC* DELs are shown on the left (**A**), and BCC tissues with *APC* DELs are shown on the right (**B**). Genes are arranged on the *x*-axis by expression level, and the log_2_-transformed gene count per million (CPM) is shown on the *y*-axis. Gene symbols for all the genes could not be shown on the *x*-axis.

**Figure 5 cells-14-00095-f005:**
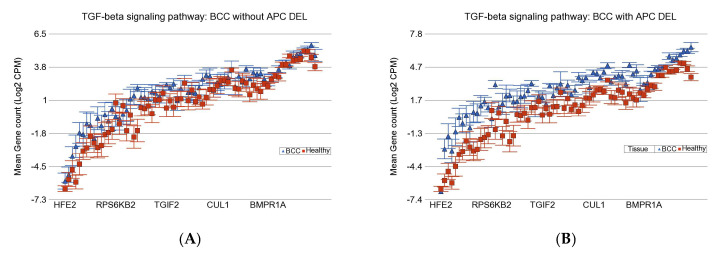
Differential gene expression of *TGF-β* pathway in BCC tissue (in blue) compared to non-lesional skin tissue (in red). BCC tissues with no *APC* DELs are shown on the left (**A**), and BCC tissues with *APC* DELs are shown on the right (**B**). Genes are arranged on the *x*-axis by expression level, and the log_2_-transformed gene count per million (CPM) is shown on the *y*-axis. Gene symbols for all the genes could not be shown on the *x*-axis.

**Figure 6 cells-14-00095-f006:**
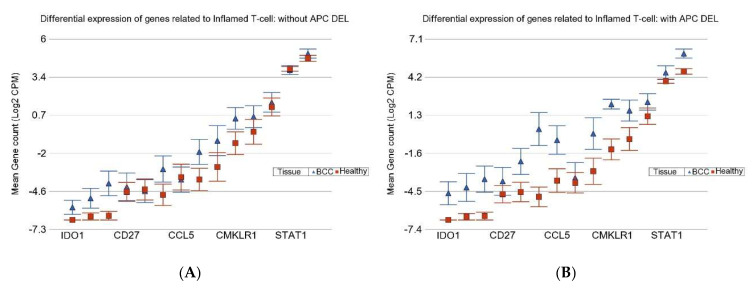
Differential expression of inflamed T-cell genes in BCC tissue (in blue) compared to healthy skin tissue (in red). In the upper panel, BCC tissues with no *APC* DELs are shown on the left (**A**) and BCC tissues with *APC* DELs are shown on the right (**B**). In the lower panel, BCC tissues from patients with high As exposure are shown on the left (**C**), and BCC tissues from patients with low As exposure are shown on the right (**D**). Genes are arranged on the *x*-axis by expression level, and the log_2_-transformed gene count per million (CPM) is shown on the *y*-axis. Gene symbols for all the genes could not be shown on the *x*-axis.

**Table 1 cells-14-00095-t001:** BCC-associated DELs in the *APC* gene. The changes are shown in multiple sources.

Coordinate	Type	Reference	Allele	Average Quality	Exact Match	Coding Region Change	Amino Acid Change
chr5:112111310	Deletion	A	-	37.00	clinvar_20210828_hg19	NM_000038.5:c.423-4delA; NM_001127510.2:c.423-4delA; NM_001127511.2:c.453-4delA	
chr5:112164717	Deletion	T	-	35.29		NM_000038.5:c.1743+55delT; NM_001127510.2:c.1743+55delT; NM_001127511.2:c.1689+55delT	
chr5:112178312	Deletion	A	-	35.29		NM_000038.5:c.7023delA; NM_001127510.2:c.7023delA; NM_001127511.2:c.6969delA	NP_000029.2:p.Lys2341fs; NP_001120982.1:p.Lys2341fs; NP_001120983.2:p.Lys2323fs
chr5:112164717	Deletion	T	-	36.48		NM_000038.5:c.1743+55delT; NM_001127510.2:c.1743+55delT; NM_001127511.2:c.1689+55delT	
chr5:112111310	Deletion	A	-	36.54	clinvar_20210828_hg19	NM_000038.5:c.423-4delA; NM_001127510.2:c.423-4delA; NM_001127511.2:c.453-4delA	
chr5:112111310..112111312	Deletion	AAA	-	33.00		NM_000038.5:c.423-6_423-4delAAA; NM_001127510.2:c.423-6_423-4delAAA; NM_001127511.2:c.453-6_453-4delAAA	
chr5:112163028	Deletion	T	-	37.00		NM_000038.5:c.1548+91delT; NM_001127510.2:c.1548+91delT; NM_001127511.2:c.1494+91delT	
chr5:112163028	Deletion	T	-	37.00		NM_000038.5:c.1548+91delT; NM_001127510.2:c.1548+91delT; NM_001127511.2:c.1494+91delT	
chr5:112164717	Deletion	T	-	35.50		NM_000038.5:c.1743+55delT; NM_001127510.2:c.1743+55delT; NM_001127511.2:c.1689+55delT	
chr5:112164717	Deletion	T	-	35.67		NM_000038.5:c.1743+55delT; NM_001127510.2:c.1743+55delT; NM_001127511.2:c.1689+55delT	
chr5:112164717	Deletion	T	-	37.00		NM_000038.5:c.1743+55delT; NM_001127510.2:c.1743+55delT; NM_001127511.2:c.1689+55delT	
chr5:112164717	Deletion	T	-	36.11		NM_000038.5:c.1743+55delT; NM_001127510.2:c.1743+55delT; NM_001127511.2:c.1689+55delT	

**Table 2 cells-14-00095-t002:** Interaction of *APC* DELs in BCC and As exposure for differential gene expressions of several pathways in NMSC compared to corresponding healthy tissue.

			*TGF-β* Signaling Pathway		*IL-17* Signaling Pathway		*PD-L1* Expression and *PD-1* Checkpoint Pathway in Cancer	
UACR			*APC* DEL − ve	*APC* DEL + ve	*APC* DEL − ve	*APC* DEL + ve	*APC* DEL − ve	*APC* DEL + ve
	>192 µg/g	FC	1.3	2.33	1.13	1.86	−1.03	2.02
		95% CI	(1.12 to 1.52)	(1.93 to 2.81)	(−1.05 to 1.34)	(1.52 to 2.29)	(−1.20 to 1.12)	(1.68 to 2.41)
		BCC Cases	*n* = 11	*n* = 6	*n* = 11	*n* = 6	*n* = 11	*n* = 6
	≤192 µg/g	FC	3.91	5.62	5.06	6.22	3.3	4.86
		95% CI	(3.21 to 4.77)	(4.53 to 6.98)	(4.06 to 6.30)	(4.90 to 7.89)	(2.72 to 4.00)	(3.94 to 5.99)
		BCC Cases	*n* = 5	*n* = 4	*n* = 5	*n* = 4	*n* = 5	*n* = 4
Interaction p			3.44 × 10^−44^		8.11 × 10^−57^		5.29 × 10^−58^	

**Table 3 cells-14-00095-t003:** Differential expression of genes associated with inflamed T-cells by *APC* DEL status and As exposure.

			Genes Related to Inflamed T-Cells	
UACR			*APC* DEL − ve	*APC* DEL + ve
	≥192 µg/g	FC	1.2	5.84
		95% CI	(−1.29 to 1.85)	(3.48 to 9.79)
		BCC Cases	*n* = 11	*n* = 6
	≤192 µg/g	FC	5.66	8.88
		95% CI	(2.97 to 10.79)	(4.48 to 17.60)
		BCC Cases	*n* = 5	*n* = 4
Interaction p			4.49 × 10^−15^	

**Table 4 cells-14-00095-t004:** Differential expression of genes associated with inflamed T-cell by *APC* DEL status and *PTCH1* ns-SNV status.

		Genes Related to Inflamed T-cell	
		*APC* DEL − ve	*APC* DEL + ve
*PTCH1* SNV − ve	FC	1.59	6.27
	95% CI	(1.05 to 2.40)	(2.77 to 14.20)
	BCC Cases	*n* = 12	*n* = 2
*PTCH1* SNV + ve	FC	6.04	7.51
	95% CI	(3.28 to 11.10)	(4.68 to 12.04)
	BCC Cases	*n* = 4	*n* = 8
Interaction p		1.79 × 10^−9^	

## Data Availability

All the supporting data are presented in the tables in the main manuscript and Appendix A.

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
