# Peer review of "Gene–Environment Interaction: Small Deletions (DELs) and Transcriptomic Profiles in Non-Melanoma Skin Cancer (NMSC) and Potential Implications for Therapy"

_cells, 2025, doi:10.3390/cells14020095_

Round 1

Reviewer 1 Report

Comments and Suggestions for Authors

The research paper is devoted to the transcriptomic analysis of the skin samples from the patients with non-melanoma skin cancer with the history of arsenic exposure monitored by non-cancer skin lesions. The study is of interest for the broad range of readers including both clinicians and fundamental researcher. However, there are several issues to be addressed before the further processing of the paper.

1)     The extensive English editing is needed: multiple typos, repeats, unfinished sentences, conversational and hyperbolic expressions, wrongly used phrases and word combination (for examples, line 320 “differential expression of inflamed T-cells”, multiply used “biologic therapy” (did the authors mean immunotherapy?), “They identified CpGs 56 showing a putative association with As exposure” (probably CpG methylation profile?)).

2)    In the Introduction section authors discussed the carcinogenicity of arsenic and its potential mechanisms. It is recommended to study and discuss the publications of regulatory committees of carcinogenic hazard such as AACR publication, IARC monographs etc.

3)    Overall formatting differs from MDPI requirements in several sections/paragraphs and should be overall checked.

4)    The abbreviation list is missing.

5)    The scheme of the study is needed: the number of patients and healthy volunteers, the analysis undertaken, the number of patients with specific DEL vs the number of the healthy volunteers. Currently it is difficult to find out the total number of patients/samples.

6)    Discussion section is too long and should be better organized. For example, authors could not describe the studies from other research groups in details but discuss it briefly and summarize in the additional table.

7)    Authors should provide the registration number for each clinical trial mentioned in the manuscript.

8)    The conclusion section should be also better organized: the authors made the conclusions without any mention of arsenic exposure of the studied patients.

Comments on the Quality of English Language

The extensive English editing is needed: multiple typos, repeats, unfinished sentences, conversational and hyperbolic expressions, wrongly used phrases and word combination (for examples, line 320 “differential expression of inflamed T-cells”, multiply used “biologic therapy” (did the authors mean immunotherapy?), “They identified CpGs showing a putative association with As exposure” (probably CpG methylation profile?)).

Reviewer 2 Report

Comments and Suggestions for Authors

The proposed study identified NMSC-associated small deletions (DEL) and the existence of gene (absence of presence of DELs)-environment (i.e. low or high As-exposure) interactions. In particular, the authors claimed significant gene-environment interactions as shown in Table 2. However, the statistical methods the authors applied to make these conclusions are problematic. Gene-environment interaction studies among other large scale cancer genomics studies are of the high-dimensionality nature (See Zhou et al. 2021). Identification of important gene-environment interactions needs to take the large scale of the cancer genomics data into account, which has not been properly handled in the current study.

Specifically, the statistical models adopted for analysis, including ANOVA and gene-set ANOVA, are only suitable for low-dimensional omics data. The gene-environment interactions shown in Table 2 should be examined within a much larger context by investigating interactions with majority (if not all) of the pathways and correct p-values for multiple-testing. It is not convincing if you only picked 3 pathways and claim they are involved in significant interaction and support the major conclusions of the paper.

Variable selection has been widely developed for gene-environment interactions. Especially recently, robust Bayesian variable selection methods have been proposed to identify important G-E interactions with inferential guarantees including confidence intervals. So compared with variable selection methods that can investigate interactions utilizing all the omics data, what’s the advantage of the low-dimensional statistical methods used in this study?

Another concern is that this study is merely based on a limited sample size around 30 along with such a large number of genomics features. So it appears an under-powered study. Have you conducted a power analysis using the current data? How can you obtain a p-value on the order of 10^(-50)? This usually happens when model assumption of ANOVA is violated and many false positive signals are generated. So, could you check if the model assumption of ANOVA is violated and examine p-values through large scale multiple testing adjustment?

References:

Zhou, F., Ren, J., Lu, X., Ma, S., & Wu, C. (2021). Gene–environment interaction: A variable selection perspective. Epistasis: Methods and Protocols, 191-223.

Round 2

Reviewer 1 Report

Comments and Suggestions for Authors

Few more comments:

1. lines 209-210. Sentence "ANOVA 209 generated p-values, correspondibf F-ratio were reported" needs correction.

2. Figure 1, Figure 4, Figure 5, Figure 6 should be given in better resolution.

Reviewer 2 Report

Comments and Suggestions for Authors

Thank you for the revision. No further comments.

Author Response

Please see the accached file
